# Refined Nomogram Incorporating Standing Cough Test Improves Prediction of Adjustable Trans-Obturator Male System (ATOMS) Success to Treat Post-Prostatectomy Male Stress Incontinence

**DOI:** 10.3390/jpm12010094

**Published:** 2022-01-12

**Authors:** Juan F. Dorado, Javier C. Angulo

**Affiliations:** 1PeRTICA Statistical Solutions, Av. Leonardo Da Vinci, 8, OF217, Getafe, 28906 Madrid, Spain; jfdorado@pertica.es; 2Clinical Department, Faculty of Biomedical Science, Universidad Europea de Madrid, Carretera de Toledo, Km 12.500, Getafe, 28905 Madrid, Spain; 3Urology Service, Hospital Universitario de Getafe, Carretera de Toledo, Km 12.500, Getafe, 28905 Madrid, Spain

**Keywords:** male stress urinary incontinence, adjustable transobturator male system, predictive nomogram

## Abstract

(1) Background: The adjustable transobturator male system (ATOMS) device serves to treat post-prostatectomy incontinence, as it enhances residual urinary sphincteric function by dorsal compression of the bulbar urethra. We investigated the clinical parameters affecting continence recovery using this device and developed a decision aid to predict success. (2) Methods: We reviewed consecutive men treated with first-time ATOMS for post-prostatectomy incontinence from 2014 to 2021 at our institution. Patient demographics, reported pads per day (PPD), 24-h pad-test and Standing Cough Test (SCT), results’ grades 1–4, according to Male Stress Incontinence Grading Scale (MSIGS), and the International Consultation on Incontinence Questionnaire-Short Form (ICIQ-SF) questionnaire were assessed. Treatment success was defined as no pads or a single PPD with ≤20-mL 24-h pad-test. Logistic regression was performed using a stepwise model (entry 0.15 and stay criterium 0.1) to evaluate independent variables’ determinant of dryness. Receiver-operating characteristic (ROC) curves for predictive variables were evaluated and their area under curve (AUC) was compared. A nomogram was generated and internally validated to predict probability of treatment success. (3) Results: Overall, 149 men (median age 70 years, IQR 7) were evaluated with a median follow-up of 45 months (IQR 26). Twelve patients (8%) had previous devices for incontinence, and 21 (14.1%) had pelvic radiation. Thirty-five men (23.5%) did not achieve continence after ATOMS adjustment (use of no or one security PPD with ≤20-mL 24-h pad-test). In univariate analysis, Charlson comorbidity index (*p* = 0.0412), previous urethroplasty (*p* = 0.0187), baseline PPD (*p* < 0.0001), 24-h pad-test (*p* < 0.0001), MSIGS (*p* < 0.0001), and ICIQ-SF questionnaire score (*p* < 0.0001) predicted ATOMS failure. In a multivariable model, 24-h pad-test (*p* = 0.0031), MSIGS (*p* = 0.0244), and radiotherapy (*p* = 0.0216) were independent variables, with AUC 0.8221. The association of MSIGS and 24-h pad-test was the superior combination (AUC 0.8236). A nomogram to predict the probability of ATOMS failure using the independent variables identified was proposed. (4) Conclusions: Several variables were identified as predictive of success for ATOMS using clinical history, physical examination (MSIGS), and factors that evaluate urine loss severity (PPD, 24-h pad-test, and ICIQ-SF questionnaire). MSIGS adds prognostic value to 24-h pad-test in assessing success of ATOMS device to treat post-prostatectomy incontinence. A nomogram was proposed to calculate the risk of ATOMS failure, which could be of interest to personalize the decision to use this device or not in the individual patient.

## 1. Introduction

Male stress urinary incontinence (SUI) after prostate cancer treatment is a very disturbing complication of this highly prevalent disease that accounts for approximately 15% of all cancers worldwide [1]. Since 2012, the adjustable trans-obturator male system (ATOMS) is increasingly used for the surgical treatment of moderate to severe male SUI [2,3].

Compared to the artificial urinary sphincter (AUS), ATOMS does not need patient manipulation, a certain advantage as cognitive and functional considerations, such as manual dexterity, affect the natural history of AUS, especially in the older population [4]. Other advantages of ATOMS over AUS include lower risk of urethral atrophy, urethral erosion, and device infection, and, also, the possibility of postoperative adjustment [5,6,7]. The mode of action of ATOMS is based on the ventral compression of the bulbar urethra, exerted by a silicone cushion that can be filled both intraoperatively and postoperatively [8]. Additionally, although several studies confirm that ATOMS is less effective in radiated patients, it can be safely used in the radiated population [9,10], which is not advisable in other fixed male slings [11].

The only randomized, controlled trial comparing fixed male sling versus AUS revealed the incontinence rate remains high after surgery, but also that both devices are useful to improve symptoms and quality of life [12]. Additionally, the DOMINO multicentric database has shown that the functional results of AUS and adjustable slings are comparable with proper preoperative patient assessment and selection [13].

Different factors can be assumed to contribute to treatment outcomes for the selection of the most proper procedure for individual patients [14]. However, studies are often limited by sample size and the use of different techniques, which may not be so comparable [5,14]. In this study, we intended to evaluate clinical factors readily available in routine practice to predict results of an individual technique, so that a personalized decision to use or not to use the device can be taken after analyzing the probability of achieving an optimal therapeutic result.

## 2. Materials and Methods

### 2.1. Study Population

We reviewed a prospectively maintained, Institutional Review Board-approved database of all men who underwent primary placement of ATOMS ^®^ (Agency for Medical Innovations, A.M.I.; Feldkirch, Austria) for SUI by the same surgical team in a university hospital, between March 2014 and July 2021. Inclusion criteria were bothering SUI after radical prostatectomy, persistent after pelvic floor exercises, in patients not considered candidates for an AUS implant because of some degree of residual sphincteric activity, and at least 3 months of follow-up after ATOMS surgery. In all cases, SUI was demonstrated by the Standing Cough Test (SCT).

Prior anti-incontinence procedures, prior urethroplasty, and pelvic irradiation were not exclusion criteria. The study was conducted in accordance with the Helsinki Declaration, and all subjects provided informed consent to participate in the study.

### 2.2. Variables Investigated

Clinical parameters registered were obtained from a standardized baseline visit before surgery. Preoperative, operative, and postoperative variables were investigated. Preoperative features included demographics (age, body mass index, ASA score, Charlson index), cancer characteristics and treatments (D’Amico risk group, radiotherapy, time since prostatectomy), and previous treatments for incontinence or urethral stricture. Continence severity baseline was assessed by SCT according to Male Stress Incontinence Grading Scale (MSIGS), 24-h pad-count (PPD), 24-h pad-test (mL), and International Consultation on Incontinence Questionnaire-Short Form (ICIQ-SF) questionnaire.

Pad-count and pad-test were registered as the average value of 3 consecutive days. The SCG ensured that patients had not voided for at least 1 h and completed a series of four forceful coughs in a standing position. Leakage was confirmed at examination with SCT in all patients. The pad was held bellow the urethral meatus and every patient was graded by the examiner according to the standardized MSIGS measurement of urine loss: 1. Only delayed drops; 2. Early drops without stream; 3. Initial drops followed by delayed stream; and 4. Early and persistent stream. The self-assessed ICIQ—SF provides a brief and robust measure to assess frequency (ICIQ item-3, 0–5 score), severity of leakage (ICIQ item-4, 0–6 score), and overall impact of incontinence (ICIQ item-5, 0–10 score). ICIQ-SF score is provided by the sum of these items (0–21 score). Urodynamic data were not included in this study, although urodynamic evaluation was generally performed to rule out obstruction and predominant detrusor overactivity.

Operative variables investigated included operative time, operative and postoperative complications (Clavien–Dindo classification), early postoperative pain (Visual Analogue Scale, 0–10) and postoperative *de novo* overactive bladder symptoms, need and cause of surgical revision, system filling after complete adjustment, and number of fillings. Evaluation of continence outcomes included 24-h pad-count (PPD) and 24-h pad-test (mL) after adjustment.

### 2.3. Surgical Technique

The ATOMS consists of a tape-shaped, mesh implant with a central integrated cushion and an access port. The surgical technique followed the original description of Seweryn et al. [2]. Under spinal anesthesia, the patient is placed in the lithotomy position, with a 14-Fr Foley catheter inserted. A vertical midline perineal incision is performed to expose bulbospongiosus muscle. The mesh arms are introduced through the obturator foramen using helical tunnelers in an outside-in technique by means of a rotary movement. The implant is brought into position by pulling the sling arms, so that the cushion gently compresses the urethra ventrally and the mesh arms are secured to the central cushion under tension. Lavage with 500 mL of a 240 mg gentamicin solution is performed.

Perioperative filling is performed after venting the cushion, up to a regular atmospheric pressure (usually 8 mL filling) or a bit more. The port is placed subcutaneously in the scrotum at a readily accessible location in case postoperative adjustment is needed by serial additional filling through scrotal puncture, until continence is reached or maximum total filling of the system, 25 mL according to the manufacturer. Patients are periodically evaluated, in concert with their follow-up for prostate cancer.

### 2.4. Study Endpoints

The primary endpoint was the evaluation of effectiveness of ATOMS, defined as use of no or one safety pad/day with a 24-h pad-test ≤ 20 mL/day. The median differential pad-test, comparing urine loss after ATOMS adjustment with respects to baseline, evaluates the magnitude of effect achieved with ATOMS implantation.

Predictive variables were investigated among clinical variables registered at baseline preoperative visit, including demographics, previous medical history, prostate cancer history, PPD, 24-h pad-test, MSGIS, and ICIQ-SF questionnaire. Based on the predictive variables identified, and their relative value for prediction, we aimed to propose a nomogram to calculate the probability of achieving continence with ATOMS. This tool could be of interest to counsel the individual patient upon the probability of incontinence cure with this implant.

### 2.5. Statistical Analysis

Statistics were calculated as the median values, interquartile range (IQR), and minimum and maximum for continuous variables, and as the frequency and percent for categorical data. Differences were calculated by the Wilcoxon rank sum test for continuous variables and the Fisher exact test for categorical ones. A *p* value < 0.05 was considered significant. The correlation between MSGIS and 24-h pad-test was evaluated. Logistic regression was performed using a stepwise model (entry 0.15 and stay criterium 0.1) to evaluate preoperatively defined independent variables’ determinant of dryness (no pad or one safety pad/day with a 24-h pad-test ≤ 20 mL/day).

The association of predicted probabilities and observed responses was evaluated and area under receiver-operating characteristic (ROC) curve for the selected model and different combinations of predictive factors was calculated. The apparent and expected optimism-corrected performances of the model were calculated using the c-index of the model with internal (bootstrap iterations, 400) validation. Finally, a nomogram to predict the probability of ATOMS failure using the independent variables identified was proposed. The statistical analysis was developed using Statistical Analysis System 9.4 (SAS Institute Inc., Cary, NY, USA).

## 3. Results

### 3.1. Description of Variables Investigated

One hundred forty-one consecutive patients with ATOMS implant to treat SUI after prostate cancer treatment in a single institution were included in the study. Device used was always that with the silicone-covered, pre-attached scrotal port design. Table 1 summarizes clinical data.

All patients in this series were previously treated for local or locally advanced prostate cancer. Radical prostatectomy was performed a median of 48 months (IQR 41 months) before ATOMS implant. Additionally, 21 patients (14.1%) had pelvic radiation. According to D’Amico risk classification, prostate cancer treated was high risk in 101 patients (67.8%). Twelve patients (8%) had previous devices for incontinence, but none had been formerly implanted with ATOMS before inclusion in the study. Preoperative urodynamic study included filling cystometry and pressure flow study was performed in 86 patients (57.7%).

### 3.2. Evaluation of Continence Outcomes

In 93 patients (66%), the 24-h pad-test after adjustment was 0 mL. Defining continence use of 1 security PPD with ≤20-mL 24-h pad-test, success to treat SUI occurred in 114 patients (76.5%), while, conversely, 35 (23.5%) did not achieve continence as defined. After ATOMS implant and postoperative adjustment, a 24-h pad test was reduced to a median 0 mL (IQR 15), which gives a statistically significant reduction compared to baseline (*p* < 0.0001), which corresponded to a median differential pad-test effect of 500 mL (460 IQR) (Figure 1).

### 3.3. Preoperative Predictors of ATOMS Success

Table 2 shows demographic and preoperative characteristics of patients undergoing ATOMS placement in the series investigated, stratified by treatment success.

Table 3 presents the corresponding odds ratios and confidence interval limits for each preoperative variable predictive of failure with ATOMS implant in the univariate analysis: Charlson comorbidity index, radiation, prostate cancer risk-group, previous urethroplasty, MSGIS group, PPD, 24-h pad-test, and ICIQ-SF total value. The multivariate analysis revealed MSIGS (category 4 vs. category 1; OR 3.412 (95% C.I. 1.159–10.095); *p* = 0.0244), radiotherapy for prostate cancer treatment (yes vs. no; OR 4.186 (95% C.I. 1.225–14.472); *p* = 0.0216), and 24-h pad-test (>1300 mL vs. ≤900 mL; OR 21.288 (95% C.I. 2.93–443.628), and 900–1300 mL vs. ≤900 mL; OR 5.591 (95% C.I. 1.802–17.903); *p* = 0.0171) stayed as independent predictive factors of failure with ATOMS implant (Figure 2).

The accuracy of the predictive model defined by the combination of the independent variables of ATOMS failure (radiation, MSIGS, and 24-h pad-test) was 82.21%. The area under the curve for MSIGS and pad-test (82.36%) was superior to that of pad-test alone (77.43%), although the difference did not reach statistical significance (ROC contrast estimation, *p* = 0.06) (Figure 3).

### 3.4. Correlation between MSIGS and 24-h Pad-Test

The Spearman’s coefficient between MSIGS and pad-test was ρ = 0.76 (*p* < 0.0001), thus demonstrating a strong positive correlation (Figure 4). With each overall increase in MSIGS grading there was a relative increase on the average 24-h pad-test. Median (IQR) pad-test value was 180 (150) mL for patients with MSIGS = 1, 250 (70) mL for MSIGS = 2, 420 (293) mL for MSIG = 3, and 900 (530) mL for MSIGS = 4 (Figure 4). This gives an idea on how MSIGS assessed by SCT is a rapid and reliable estimate of incontinence severity.

### 3.5. Nomogram Generation and Internal Validation

Using the multivariate logistic regression model presented a nomogram can be generated to predict the overall probability of ATOMS failure in the particular patient (Figure 5). This model was internally validated by bootstrapping with 82.2% (95% CI 81.7–82.7) apparent performance and 11.1% (95% CI 10.7–11.5) expected optimism.

## 4. Discussion

The selection of the ideal implant for the patient with moderate to severe male SUI is complicated even in centers of expertise. When offering both AUS and adjustable slings, the decision is mainly based on considering the history of radiation therapy and previous failed incontinence therapy [13]. Despite the scarcity of direct comparative studies, more complex patients are generally selected for an AUS implantation than for other options, and that may have a possible impact on the postoperative outcome [5,13]. Still though, the functional results and satisfaction with incontinence devices may be comparable [5,12].

ATOMS is probably the adjustable sling with a larger accumulated body of evidence both regarding effectiveness and safety, especially compared to other adjustable continence systems for male SUI, such as Pro-ACT and Male Remeex System [3,15,16]. ATOMS is widely used in Europe and Canada as an alternative to AUS [9,10,17,18] and is being currently evaluated by the FDA.

It can be a matter of debate whether adjustable slings, and specifically ATOMS, should only be used in patients with mild to moderate incontinence or if it can be used to treat selected patients with severe incontinence as well, taking into account that patients with total sphincteric deficiency are not good candidates for ATOMS and would be better treated with an AUS [8]. In this respect, patients with age-related cognitive decline are better treated with an ATOMS as no manipulation is needed for micturition with the adjustable device. Additionally, although the theoretical margin of SUI severity improvement (pad-test change) for AUS exceeded that of ATOMS, the revision rate seems much higher for AUS [5,6]. Taking all this into account, it can be understandable that patient satisfaction after an ATOMS implant can be high, even when cases with severe SUI are included [19].

As was recently demonstrated, the SCT can help better stratify moderate male SUI (MSIGS 0–2 vs. 3–4) to more accurately predict sling success [20]. On the other hand, the 24-h pad-test provides a reliable and objective assessment of continence rates in patients with an AUS and strongly correlates to the ICIQ-SF score, so as to reduce reported outcome heterogeneity across studies [21]. However, some patients find it difficult and tedious to collect and weigh pads for a full 24-h period, and also the variation of conditions from one day to another in water intake and exercise limits the applicability of the pad-test to make the decision whether to choose one or another technique. One can assume fixed slings can be used to treat the less severe cases, AUS the most severe ones, and adjustable slings have a better chance for those in between [22,23]. In this context, the SCT could have a role to better select candidates as it is a straightforward examination easily available in the clinic [24].

Many options have been suggested to ease the limitations of the 24-h pad-test. Use of a 1-h pad-test is an attractive option to facilitate the evaluation, but, despite the need for standardization of the test, it may not overcome the limitations of the 24-h evaluation [25]. On the other hand, a 7-day pad-test has been proposed as another alternative with added value in SUI after prostatectomy but is much more cumbersome for the patient [26]. Another strategy to evaluate incontinence severity is the self-administered ICIQ-SF questionnaire, a robust and also straightforward method to assess the impact of post-prostatectomy SUI [27], which can be used in conjunction with the pad count [28,29].

We confirmed that the modified SCT assessed by MSIGS is a rapid and reliable estimate of incontinence severity, as was originally demonstrated by Yi et al., based on the strong correlation between SCT, the 24-h pad-test, and the patient-reported PPD [30]. Additionally, for patients with detrusor overactivity and mixed incontinence, a pad-test may be less reliable than a cough (Valsalva-based) assessment [31]. Additionally, the incorporation of SCT to predictive models of sling success was confirmed to improve patient selection for fixed male transobturator sling [32].

Previous attempts to assess postoperative outcomes from the ATOMS and identify factors influencing failure to achieve continence agree that concurrent radiotherapy and increased pre-operative pad usage are independent factors associated with failure to achieve continence [10,33]. With the intention to improve patient selection for candidates to adjustable transobturator male sling ATOMS, we developed a clinical tool to aid in preoperative patient evaluation and counseling. Urodynamics can help in the selection of optimal candidates for ATOMS by evaluation of preoperative voiding phase [34]. However, a more reproducible testing based on clinical preoperative variables is needed to improve the prediction of surgical failure. A nomogram based on a 24-h pad-test, SCT graded by MSIGS, and radiation previous history could be used to clinically support the decision of implanting ATOMS or AUS, especially in cases with moderate–severe urine loss. We did not include data of preoperative voiding phase by urodynamics in the multivariate analysis because they were not available in all the patients investigated.

The main limitation for this multivariate analysis and model development stands on the fact that it has been based on single-institution data, and external validation is advisable so that it can be generalized. Additionally, the definition of failure is rather stringent and does not consider patient-reported outcomes. The main strength of the model is that it is based on simple measurements. Patients without radiation, with a baseline 24-h pad-test ≤ 900 mL, and with SCT other than early and persistent stream are the best candidates to consider ATOMS implant. In fact, a pad-test median variation of 500 ± 460 mL has been confirmed as a magnitude effect for ATOMS implant in this series, higher than the 200–400 mL that was observed after fixed male sling implant [35,36].

## 5. Conclusions

In men with post-prostatectomy stress incontinence considered candidates for ATOMS implantation, quantification of the SCT seems a non-invasive and rapid assessment of incontinence severity, which correlates well with the 24-h pad-test. Outcomes of ATOMS regarding dryness after adjustment (use of no or one safety pad/day with a 24-h pad-test ≤ 20 mL/day) can be predicted with a simple nomogram incorporating SCT (MSGIS scale), 24-h pad-test, and history of radiation. An ideal candidate for an ATOMS has baseline 24-h pad-test ≤ 900 mL, SCT grades 1–3, and no history of radiation. This clinical tool was confirmed by bootstrap resampling but needs validation in an external cohort.

## Figures and Tables

**Figure 1 jpm-12-00094-f001:**
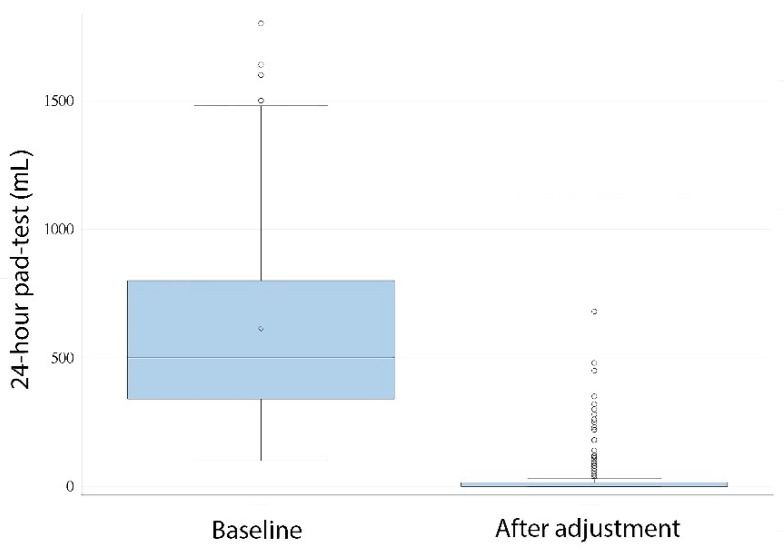
The 24-h, pad-test baseline compared to after adjustment (*t*-test, *p* < 0.0001).

**Figure 2 jpm-12-00094-f002:**
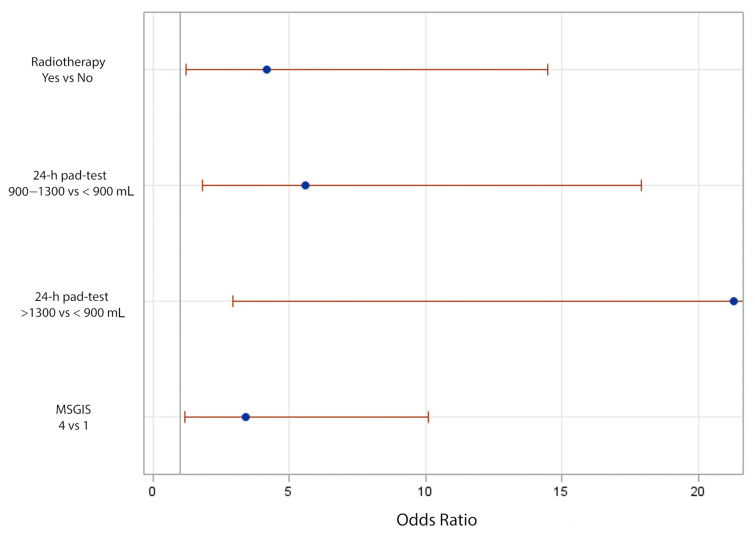
Odds ratios with 95% profile-likelihood confidence limits.

**Figure 3 jpm-12-00094-f003:**
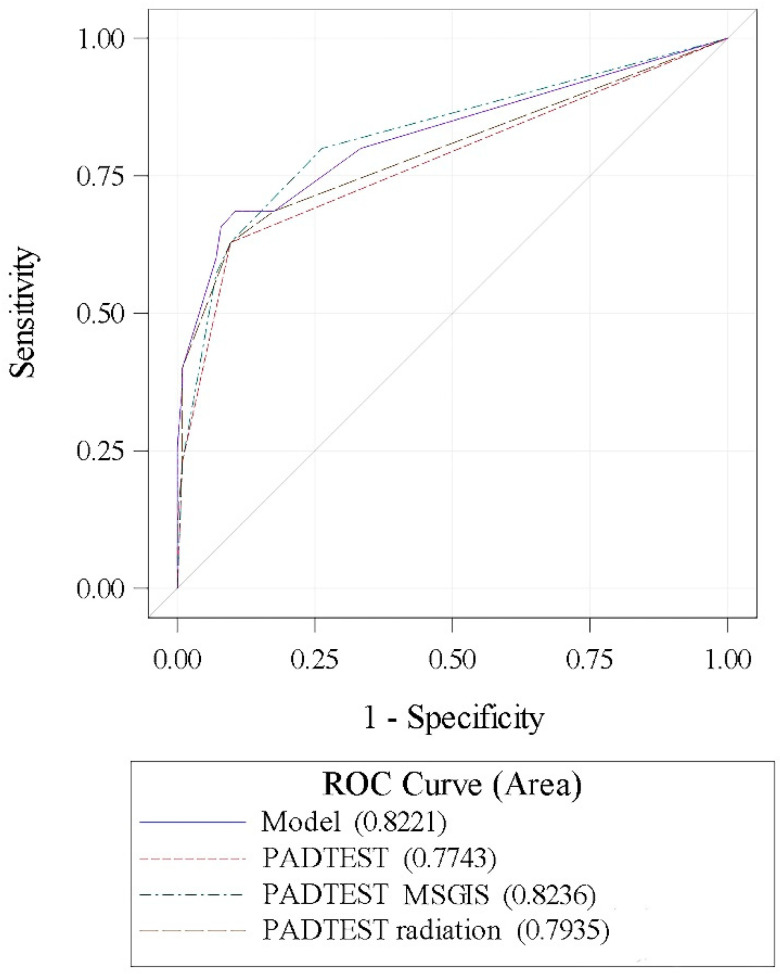
ROC curves for comparisons.

**Figure 4 jpm-12-00094-f004:**
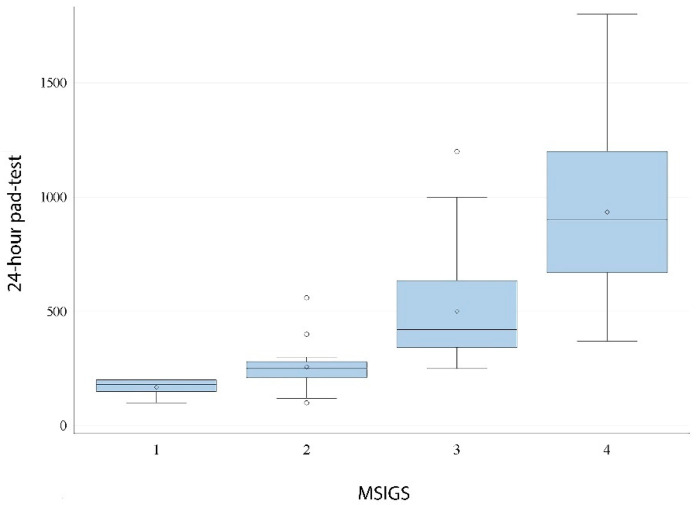
Distribution of 24-h pad-test by MSIGS (Spearman’s ρ = 0.76; *p* < 0.0001).

**Figure 5 jpm-12-00094-f005:**
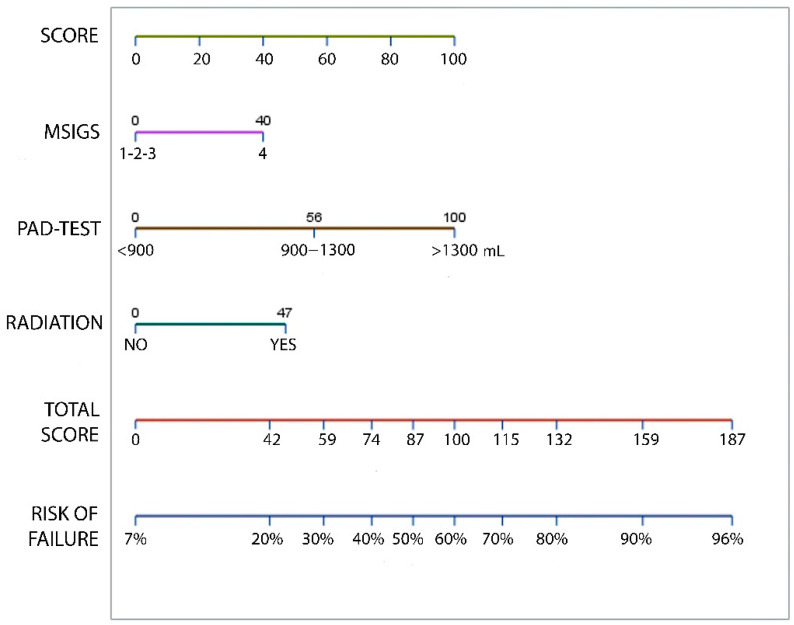
Nomogram predicting the probability of ATOMS failure, which can be calculated by obtaining the value for each parameter by drawing a straight line to the point axis, adding the points together, and finding the sum on the total point axis.

**Table 1 jpm-12-00094-t001:** Preoperative, operative, and postoperative data of patients included (*n* = 141).

Variable	*n* (%)
**Preoperative data**	
Age, years, median (IQR, range)	70 (7, 49–83)
Body mass index, median (IQR, range)	26.6 (4.8, 16.2–40.2)
ASA score, *n* (%)	
ASA I category	27 (18.1)
ASA II category	97 (65.1)
ASA III category	25 (16.8)
Charlson comorbidity index, median (IQR, range)	4 (2, 1–9)
Previous incontinence surgery, *n* (%)	12 (8%)
Previous urethroplasty, *n* (%)	11 (7.4)
Previous radiation, *n* (%)	21 (14.9)
D’Amico prostate cancer risk group ^(1)^, *n* (%)	
Low risk	16 (10.7)
Intermediate risk	32 (21.5)
High risk	101 (67.8)
Time since prostatectomy, months, median (IQR, range)	48 (41, 11–160)
Male Stress Incontinence Grading Scale (MSIGS) ^(2)^, *n* (%)	
MSIGS 1 ^(2)^	6 (4)
MSIGS 2 ^(2)^	17 (11.4)
MSIGS 3 ^(2)^	73 (49)
MSIGS 4 ^(2)^	53 (35.6)
24-h pad count (PPD) ^(3)^, *n*, median (IQR, range)	5 (3, 1–11)
24-h pad test, mL, median (IQR, range)	500 (460, 100–1800)
ICIQ-SF total, median (IQR, range)	15 (5, 9–21)
ICIQ-SF Question 1	4 (0, 3–5)
ICIQ-SF Question 2	4 (2, 2–6)
ICIQ-SF Question 3	6 (3, 3–10)
**Operative data**	
Operative time, min, median (IQR, range)	55 (22, 25–135)
Perioperative complication, *n* (%)	2 (1.4)
Postoperative complications ^(4)^, any grade, *n* (%)	31 (22)
Grade I ^(4)^, *n* (%)	23 (16.3)
Grade II ^(4)^, *n* (%)	2 (1.4)
Grade III ^(4)^, *n* (%)	6 (4.3)
VAS for pain (0–10), median (IQR, range) ^(5)^	0 (1, 0–8)
**Postoperative data**	
Total filling volume, mL, median (IQR, range)	15 (8, 8–37)
Number of fillings, *n*, median (IQR, range)	1 (3, 0–7)
Follow-up since implant, months, median (IQR, range)	45 (26, 6–89)
Patients with pad-test ≤ 20 mL, *n* (%)	114 (76.5)
Patients with pad-test zero mL, *n* (%)	93 (66)
24-h pad count (PPD), *n*, median (IQR, range)	0 (1, 0–6)
24-h pad test, mL, median (IQR, range)	0 (15, 0–680)
Differential 24-h pad test ^(6)^, mL, median (IQR, range)	500 (460, 30–1600)

^(1)^ Before prostate cancer therapy; ^(2)^ MSIGS according to standing cough test; ^(3)^ PPD, pads-per-day; ^(4)^ According to Clavien–Dindo classification; ^(5)^ At discharge, usually on day 1 after surgery; ^(6)^ Baseline minus after adjustment for 24-h pad-test, expressed in mL (magnitude change). ASA, American Society of Anesthesiologists; ICIQ-SF, International Consultation on Incontinence Questionnaire-Short Form.

**Table 2 jpm-12-00094-t002:** Variables stratified by treatment success.

Parameter	Success		*p* Value
	Yes (*n* = 114)	No (*n* = 35)	
Patient age at implantation	70 (66–73)	71 (66–73)	0.826
BMI at implantation	26.3 (24–28.3)	27.1 (24.8–29.7)	0.0876
Charlson comorbidity index	4 (3–5)	5 (4–6)	0.0376
Previous urethroplasty	5 (4.4%)	6 (17.1%)	0.021
Previous incontinence device	12 (10.5)	0 (0%)	0.0694
Radiotherapy	9 (7.9%)	12 (34.3%)	0.0003
Intermediate-risk group	29 (25.4%)	3 (8.6%)	0.0713
Time since prostatectomy	48 (36–77)	52 (32–81)	0.8332
Pads per day	4 (3–6)	8 (6–8)	<0.0001
24-h pad-test	425 (300–670)	950 (600–1200)	<0.0001
MSIGS 4	27 (23.7%)	26 (74.3%)	<0.0001
ICIQ-SF	14 (13–18)	18 (14–21)	<0.0001

Continuous variables are presented as medians with interquartile ranges in parenthesis. MSIGS, Male Stress Incontinence Grading Scale; ICIQ-SF, International Consultation on Incontinence Questionnaire-Short Form.

**Table 3 jpm-12-00094-t003:** Logistic regression model to predict ATOMS failure.

**Univariate Analysis**	**Odds Ratio**	**95% CI**	***p*-Value**
Charlson comorbidity index (2 vs. 1)	2.717	1.101–7.752	0.0412
Radiotherapy (Yes vs. No)	6.087	2.317–16.605	0.0003
Prostate cancer risk (Intermediate vs. Low)	2.817	1.154–8	0.0339
Previous incontinence surgery (Yes vs. No)	0.001	1.515–	0.9617
Previous urethroplasty (Yes vs. No)	4.504	1.274–16.666	0.0187
MSIGS group (4 vs. 1)	9.309	4.02–23.341	<0.0001
PPD (per unit)	1.826	1.448–2.39	<0.0001
24-h pad-test (>1300 vs. ≤900 mL)	63.385	10.474–999	<0.0001
24-h pad-test (900–1300 vs. ≤900 mL)	11.092	4.178–31.056	
ICIQ-SF (per unit)	1.369	1.196–1.588	<0.0001
**Multivariate Analysis**	**Odds Ratio**	**95% CI**	***p*-Value**
MSIGS group (4 vs. 1)	3.412	1.159–10.095	0.0244
Radiotherapy (Yes vs. No)	4.186	1.225–14.472	0.0216
24-h pad-test (>1300 vs. ≤900 mL)	21.288	2.93–443.628	0.0171
24-h pad-test (900–1300 vs. ≤900 mL)	5.591	1.802–17.903	

## Data Availability

Full data will be provided by the corresponding author upon reasonable request.

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
