# Peer review of "Refined Nomogram Incorporating Standing Cough Test Improves Prediction of Adjustable Trans-Obturator Male System (ATOMS) Success to Treat Post-Prostatectomy Male Stress Incontinence"

_jpm, 2022, doi:10.3390/jpm12010094_

Round 1

Reviewer 1 Report

Comments to authors

This study evaluated the role of standing cough test (SCT) in the prediction of success of adjustable trans-obturator male system (ATOMS) in incontinent patients after prostatectomy, when it has incorporated in a predictive nomogram. This article, which is a single center research, is trying to point out the importance of the standing cough test for the choice therapy in patients with post prostatectomy incontinence. Indeed, it is the first analysis regarding this subject, which includes the SCT in a predictive nomogram for the success of ATOMS device in these patients. Length and readability are good. The choice of topic is interesting because the success of treatments in post prostatectomy incontinent patients is a big issue. The only drawback, pointed by the authors, is that it is a single institute analysis and the data, should be validated from other studies.

My opinion is that this paper would be possible accepted without any revisions

Author Response

Thank you very much for the comments.

Reviewer 2 Report

This work fits in the wake of other papers by the same authors and overall it represents an interesting and timely study which attempts to provide useful practical information to investigate the clinical parameters affecting continence recovery using the adjustable transobturator male system (ATOMS) device to treat post-prostatectomy incontinence. The authors developed a nomogram to calculate the risk of ATOMS failure, that could help to personalize the decision to use this device or not in the individual patient. I think that this work could be improved adding evaluation of preoperative voiding phase by urodynamics to have a more reproducible testing in the selection of optimal candidates for ATOMS and to improve the prediction of surgical failure. Nevertheless, I think that this study proposes an interesting approach to the task and the readability of the paper is enough to read.

Author Response

Thank you very much for the comments.

A revised version of the manuscript includes the number of patients with preoperative urodynamics performed, that makes a 58% of the total numebr of patients included in the stydy. For this reason the urodynamic data were not included in the logistic regression analysis.

Two new sentences are included, both in material and methods  (lines 174-176), and in the discussion (lines 295-296) sections. To address this issue.